# Comparison of Automatic Classification Methods for Identification of Ice Surfaces from Unmanned-Aerial-Vehicle-Borne RGB Imagery

**Jakub Jech** [1],*[ID]**, Jitka Komárková** [1][ID] **and Devanjan Bhattacharya** [2][ID]

1   Institute of System Engineering and Informatics, Faculty of Economics and Administration,
    University of Pardubice, Studentská 95, 532 10 Pardubice, Czech Republic; jitka.komarkova@upce.cz
2   Bayes Centre, The University of Edinburgh, 47 Potterrow, Edinburgh EH8 9BT, UK; d.bhattacharya@ed.ac.uk
*   Correspondence: jakub.jech@upce.cz

**Abstract:** This article describes a comparison of the pixel-based classification methods used to distinguish ice from other land cover types. The article focuses on processing RGB imagery, as these are very easy to obtained. The imagery was taken using UAVs and has a very high spatial resolution. Classical classification methods (ISODATA and Maximum Likelihood) and more modern approaches (support vector machines, random forests, deep learning) have been compared for image data classifications. Input datasets were created from two distinct areas: The Pond Skříň and the Baroch Nature Reserve. The images were classified into two classes: ice and all other land cover types. The accuracy of each classification was verified using a Cohen's Kappa coefficient, with reference values obtained via manual surface identification. Deep learning and Maximum Likelihood were the best classifiers, with a classification accuracy of over 92% in the first area of interest. On average, the support vector machine was the best classifier for both areas of interest. A comparison of the selected methods, which were applied to highly detailed RGB images obtained with UAVs, demonstrates the potential of their utilization compared to imagery obtained using satellites or aerial technologies for remote sensing.

**Keywords:** imagery classification; RGB imagery data; UAV; supervised classification; unsupervised classification; Iso Cluster; Maximum Likelihood; random trees; support vector machine; deep learning; pixel-based classification

## 1. Introduction

Remote sensing is nowadays an essential tool for everyday needs [1], especially with the use of GIS tools, whether it is logistics, spatial planning, or landscape monitoring [2–4]. Introducing unmanned aerial vehicles (UAVs) as remote sensing carriers has led to a new era in the industry, especially for applications requiring a high spatial resolution.

Image classification is one of the essential functions of the GIS [5]. It is used to process data, and, for example, it is also an essential tool for identifying particular land cover types and monitoring their changes.

There are many methods for automatic image classification, and they can generally be divided into two groups: unsupervised automatic classification (learning without a teacher) and supervised classification (learning with a teacher).

The classification of images using automatic methods is highly required, especially for precision agriculture [6], but it is also widely applied throughout all other socio-economic and scientific fields. Along with very high spatial resolution, high accuracy for classifications to particular land cover types is necessary [7,8]. Therefore, this article deals with this topic.

Over the last ten years, the classification of image data has been a very popular method of data processing through various industries, even in scientific works. In the last five years, there has been a significant increase in the number of works dealing with the classification

of imagery data, see Figure 1. The combination of image classification, "drone", and RGB is less covered by the published works than the combination of deep learning with "drone" and RGB. The focus on particular classification methods is significantly lower, see Tables 1 and 2. The term "drone" stands for UAV, UAS, drone, RPAS, MAV, unmanned aerial vehicle, and unmanned aerial system. This low attention is another reason for research in this area.

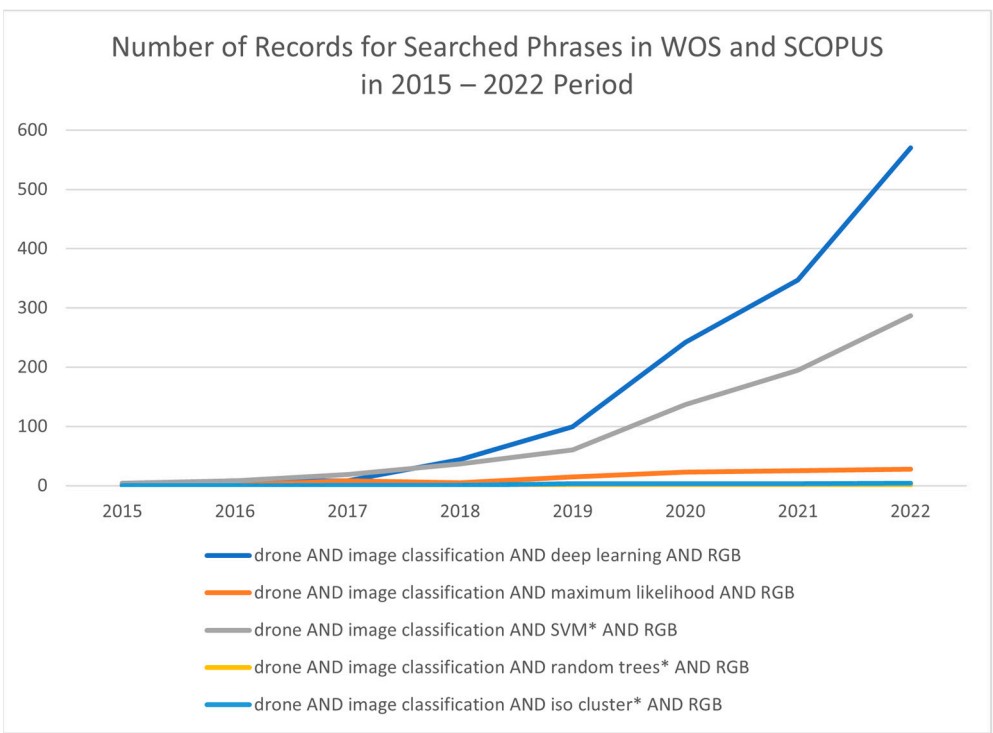

**Figure 1.** Chart of the evolution of keyword search focusing on image classification and classification methods (2015–2022).

**Table 1.** Numbers of records focusing on RGB UAV-borne data classification in WOS and SCOPUS databases (2000–2022).

| Searched Phrases in WOS and SCOPUS (2000–2022) | WOS | SCOPUS |
|---|---|---|
| image classification AND drone AND RGB | 56 | 1878 |
| deep learning AND drone AND RGB | 56 | 4885 |
| maximum likelihood AND drone AND RGB | 25 | 254 |
| random trees * AND drone AND RGB | 6 | 99 |
| iso cluster ** AND drone AND RGB | 1 | 27 |
| SVM *** AND drone AND RGB | 91 | 1873 |

The term drone represents UAV, UAS, drone, RPAS, MAV, unmanned aerial vehicle, unmanned aerial system; * random tree and random trees; ** Iso Cluster, isodata; *** SVM— support vector machine.

It is also important to mention that this article focuses on the automatic classification] of RGB image data obtained from UAVs. Most remote sensing data are obtained as multispectral or hyperspectral. Such images contain units of up to tens of spectral bands, each with its own specific spectral behaviour. The advantages of different bands are in the subsequent classification of an image; e.g., scanning in the NIR band is suitable for monitoring chlorophyll in vegetation.

The availability of UAVs is very high nowadays. They represent a non-invasive observation and data collection device. Smaller UAVs allow most users to retrieve their own data without any problems [9]. The significant advantage of UAVs is the data quality,

namely a very high spatial resolution in cm × px$^{-1}$. Smaller UAVs equipped with an RGB camera can also be considered a low-cost data acquisition solution [10]. The sensor is one of the main parameters that affect the price and quality of images.

**Table 2.** Records in WOS and SCOPUS database—summary for the terms.

| Key Word | WOS | SCOPUS |
| --- | --- | --- |
| UAV | 28,423 | 57,575 |
| drone | 8308 | 19,923 |
| UAS | 6673 | 9839 |
| Unmanned aerial vehicle | 15,480 | 53,445 |
| Unmanned aerial system | 942 | 3391 |
| RPAS | 657 | 937 |
| MAV | 2476 | 6692 |

When looking for relevant contributions in scientific databases, it is necessary to clarify the terminology [11–13] because many terms are used. UAV, or unmanned aerial vehicle, is the technical term, which is the most widely used term for these machines. A UAV is a correct designation only for the machine itself. An unmanned aircraft system (UAS) is used to describe the entire set. This designation is often used in Anglo-Saxon terms. The term drone can be found very often as well. It comes from the French designation. The term drone has gained tremendous popularity for naming machines among the public and in non-scientific contributions. The most formal and international term is remotely piloted aircraft system or RPAS [14]. With the development of technology and the downsizing of the machines themselves, the term micro aerial vehicle (MAV) can also be found.

Therefore, it is necessary to consider these terms when searching for scientific contributions. Table 2 shows the frequency of occurrences of particular terms when searching in the scientific databases WOS and SCOPUS. Figure 2 shows that it is appropriate to focus on the terms UAV, UAS, drone, and unmanned aerial system when searching. Other terms are not used so often compared to the above-stated terms. All the terms are often used in a supplementary way.

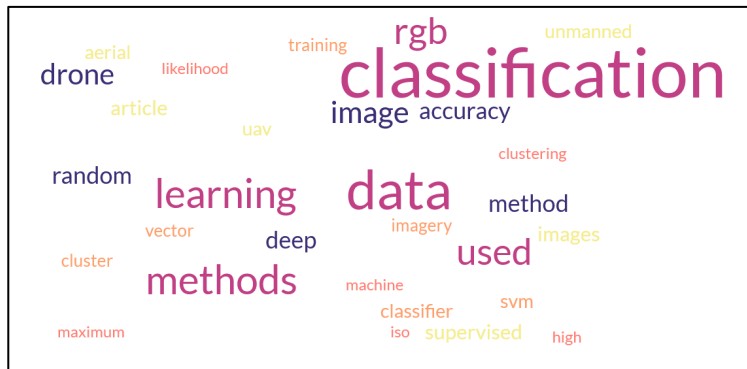

**Figure 2.** Infographics pointing out the frequency of particular terms in the article.

As mentioned earlier, image classification methods can be divided into two fundamental types of methods. The first type is an automatic image classification based on learning without a teacher or unsupervised classifications. The basis of this approach is the clustering method, i.e., it is based on the clustering of pixels based on the similarity of a given classification class. The second type is an automatic classification based on learning with a teacher or supervised classification. The basis for this approach is the creation of training sets with the correct classification outputs, i.e., the required classes. After selecting a training set, they are applied to the testing set [15].

Unsupervised classification methods are limited by only the clustering method, so a few methods for classifying image data exist. The most frequently used methods are ISODATA and the K-means method. The advantage of these methods is that they are given, as an input criterion, only the number of clusters, and then the methods run on their own.

Supervised methods are based on learning with a teacher, thus on training and test sets. The required classification classes are selected on a training set, and the selected methods are taught. Taught training sets are applied to the test set, and the output is a classified image. The disadvantage of this may be the error rate of entering training areas when overlaps of different classes may occur on the same data. The most frequently used supervised method is Maximum Likelihood method, which has been used for a long time [16]. A newer method for supervised classification is SVMs (support vector machines). Next, deep learning methods are more often used for image classification in the GIS, and they were successfully used in various studies: tree species classification using RGB imagery and deep learning [17] or classification of fluvial scenes [18] may be given as examples. An automated pipeline based on deep learning was designed to identify particular animal species [19]. The identification of plant leaf diseases is another useful application of deep learning [20].

An accuracy assessment ([21], pp. 306–308) is essential to any classification project. It compares the classified image with another data source considered accurate or ground truth data. The ground truth can be collected in the field, which is time-consuming and expensive. The ground truth data can also be derived from interpreting high-resolution imagery, existing classified imagery, or GIS data layers. The ground truth data can be provided as a reference layer or created by manual identification. This process is time-consuming and requires knowledge of the research area.

The most common way to assess the accuracy of a classified map is to create a set of random points from the ground truth data and compare that with the classified data in a confusion matrix [22]. A confusion matrix, also known as an error matrix, is a specific table layout that allows the performance of an algorithm to be visualized, typically a supervised learning one. Each row of the matrix represents the instances in an actual class, while each column represents the instances in a predicted class. A confusion matrix is a table with two rows and two columns (representing actual and predicted classes) that reports the number of false positives, false negatives, true positives, and true negatives. It allows for more a detailed analysis than a mere proportion of correct classifications (accuracy).

Output data from a confusion matrix must be represented. Cohen's Kappa index [23] measures valid agreement. It indicates the proportion of agreement beyond the one expected by chance, that is, the achieved beyond-chance agreement as a proportion of the possible beyond-chance agreement.

The Cohen's Kappa index is calculated by Equation (1), where $Po$ is the relative observed agreement among raters; $Pc$ is the hypothetical probability of chance agreement. The Kappa index is represented in the interval 0 to 1, where 1 means a maximum match.

$$K = \frac{observed\ agreement - chance\ agreement}{1 - chance\ agreement} = \frac{P_O - P_C}{1 - P_C} \tag{1}$$

The aim of this article is a comparison of selected automatic classification methods for ice detection from RGB imagery. ISODATA and Maximum Likelihood represent an older approach, and SVM, random forest, and deep learning represent a newer approach. The methods were used to classify land cover types into two classes (ice and all other land covers). Very-high-spatial-resolution UAV-borne datasets were used. The following infographics, see Figure 2, point out the importance of particular topics.

### 1.1. Iso Cluster

ISODATA computes class means consistently circulated in the data space before iteratively clustering the continuing pixels utilizing least distance approaches [24]. ArcGIS software (10.3 version) tools use this method, which is called an Iso cluster.

The Iso cluster [25] tool uses a modified iterative optimization clustering procedure known as the migrating means technique. The algorithm separates all cells into the user-specified number of distinct unimodal groups in the multidimensional space of the input bands. The Iso prefix of the ISODATA clustering algorithm is an abbreviation for the iterative self-organizing way of performing clustering. This type of clustering uses a process in which, during each iteration, all samples are assigned to existing cluster centres and new means are recalculated for every class. The optimal number of classes to specify is usually unknown. The Iso Cluster algorithm is an iterative process for computing the minimum Euclidean distance when assigning each candidate cell to a cluster ([21], pp. 297–299).

### 1.2. Maximum Likelihood

The Maximum Likelihood [16] classification assumes that the statistics for each class in each band are normally distributed and calculates the probability that a given pixel belongs to a specific class. Unless you select a probability threshold, all the pixels are classified. Each pixel is assigned to the class with the highest probability (the Maximum Likelihood). The pixel remains unclassified if the highest probability is smaller than the specified threshold.

### 1.3. Random Trees

In its simplest form, a random forest [26] can be thought of as using the bagging and the random subsets meta-classifier on a tree classifier. The random forest [27] classifier consists of a combination of tree classifiers. Each classifier is generated using a random vector, sampled independently from the input vector, and each tree casts a unit vote for the most popular class to classify an input vector. The random forest introduces randomness of two types: each tree is built on slightly different rows sampled with repetitions from the original (bagging), and each column tree (or, in some cases, each branch decision) is built using a small randomly selected subset of columns [28].

The random trees [29] is the classification method used in ArcGIS software and is based on the random forest.

### 1.4. Support Vector Machine

A support vector machine (SVM) [30] is a supervised learning model with associated learning algorithms that analyse data for classification and regression analyses. SVM is one of the most robust prediction methods based on statistical learning frameworks, or the VC theory proposed by Vapnik (1995) and Chervonenkis (1974) [30]. Given a set of training examples, each marked as belonging to one of two categories, an SVM training algorithm builds a model that assigns new examples to one category or the other, making it a non-probabilistic binary linear classifier. SVM [31] maps training examples to points in space to maximize the gap width between the two categories. New examples are then mapped into that space and predicted to belong to a category based on which side of the gap they fall into.

### 1.5. Deep Learning

Deep learning [32] is a subset of machine learning that uses several layers of algorithms in the form of neural networks. The input data are analysed through different layers of the network, with each layer defining specific features and patterns in the data. For example [33], if you want to identify features such as buildings and roads, the deep learning model can be trained with images of different buildings and roads, processing the images through layers within the neural network, and then finding the identifiers required to classify a building or road.

## 2. State of Art

The authors of one article [34] used a different approach to image classification. This work's advantage is that image classification methods were used for RGB data from a UAV. The Iso Cluster and Maximum Likelihood methods are older but widely used methods and sometimes provide good results. Images were classified into three classes in two time records. In this article, a 46% accuracy of unsupervised classification (also the worst) and 92% accuracy of supervised classification (also the best) were achieved.

The article [18] compares selected supervised classification methods to classify fluvial scenes. Data were in RGB spectre from the existing dataset. The dataset was formed by 11 rivers worldwide (Japan, Italy, Canada, the UK, and Costa Rica) in the RGB images. In the first phase, they compare Maximum Likelihood, random forest, and a multilayer perceptron with a 70–80% F1 result. In the second phase, they use the CSC method and obtain a 92–95% F1 result. This demonstrates the suitability of deep learning or methods based on deep learning.

The study [35] introduces a technique to use UAV-acquired RGB images coupled with ground information for a reliable and fast estimation of sugarcane yield for two popular varieties in Thailand. This article is interested in two first approaches (OBIA and ExG). OBIA provided better results with 92 and 96% accuracy. The pixel-based method provided 84 and 88% accuracy. The utilization of the UAV with a standard camera with an RGB sensor and the OBIA method are the contributions of this study. The OBIA method was identified as a good classifier for heterogeneous data.

The study's primary goal [36] was to measure the accuracy of the selected methods (SVM, ML) to assess historical, current, and future land use and land cover patterns. The data were based on satellite imagery (river Birupa), and the methods were compared. The data were obtained from Landsat with a 30 m spatial resolution. From the results, it is possible to observe that both classifiers in some land covers (build-up area and fallow land) have identical accuracy. Furthermore, you can see in other land cover types that the results are very different. According to Kappa statistics, the SVM is a better classifier with an 86% average accuracy.

This article [37] describes an automatic method of identifying soil and water conservation measures from the centimetre-resolution imagery of UAVs. The authors chose an object-based image analysis (OBIA) approach, machine learning models, and a support vector machine (SVM). Data were obtained from a UAV with a low-cost RGB camera. The images from a UAV provide have very high spatial resolution in cm*px$^{-1}$. After obtaining the data, they used vegetation indices as the first step. They obtained 91% accuracy.

The article [38] classifies sea ice from high-resolution observations. Data were collected during OIB Arctic summer campaigns with the nominal flight altitude. They obtained RGB images from a Canon EOS 5D camera with a high spatial resolution of 10 cm*px$^{-1}$. They are classified into selected classes by histograms. This article focused on classifying ice and water in the RGB spectre.

The article [39] investigates the potential of SAR imagery for sea ice classification. This article classified the imagery into two approaches (pixel-based and region-based). The classification accuracy of ice ranges from 80 to 90%, depending on the set of classification classes used.

The classification of sea ice images was the aim of the authors [40]. In this article, hyperspectral data obtained from satellites were used. As a classifier, deep learning was used, specifically, 3D-CNNs. They reach very high accuracy, around 98%.

The authors of another study [41] focused on differentiating snow and rock in colour imagery (RGB). They used unsupervised and supervised approaches. The Maximum Likelihood classification and a new approach, Polynomial Thresholding were used as supervised methods. Both supervised approaches reach an accuracy of around 95%.

## 3. Data Collection, Preprocessing and Methodology

The procedure used in this study consisted of the following steps:

- Data collection:
- Flight planning
- Flight itself
- Data preprocessing—mosaics creation
- Data processing—classifications
- Results visualization

The aim was to compare the used methods, but there was no specific order of the classification methods set.

### 3.1. Used Hardware and Software

Experiments, calculations, and visualization were performed on a machine with an Intel Core i7 at 2.6 GHz and 16 GB of RAM, which represents acceptable computational requirements for the used data. The only exception is deep learning. Its performance can be improved by using a GPU (Nvidia GTX 960) for computation. Using a GPU provides much more effective processing of data. The GPU calculation of the deep learning method provides faster processing, from 6 h for only a CPU-based calculation to 30 min using both CPU and GPU.

DJI GS PRO, ENVI OneButton, and ArcGIS software were used for the following: DJI GS PRO (on iOS, version 2.0.13) was used to plan and control flights, and ENVI OneButton (version 5.0.0.181) was used to create mosaics from collected images. ArcGIS for Desktop (version 10.6) and ArcGIS PRO (version 2.7) are GIS software used for image classification by unsupervised and supervised methods and for accuracy assessment.

### 3.2. Area of Interest

This study is focused on two small areas during the winter term. The first one is situated close to the village of Neratov (next to the city of Lázně Bělohrad and near Pardubice) in the Pardubický region, the Czech Republic. The north-western part of the pond Skříň was monitored (see Figure 3). The total area of the pond is 269,000 m$^2$, and the pond is used for fish farming. Pond Baroch was chosen as the second area of interest (see Figure 4). Pond Baroch is part of a nature reserve with evidence number 1926. The pond is situated south to southwest of the village of Hrobice, near Pardubice. The Regional Office of the Pardubice Region manages the area. The size of the nature reservation is around 30,000 m$^2$. The reason to protect this nature reserve is its grounded pond, adjacent reeds, forest and meadow communities, and ornithological locality.

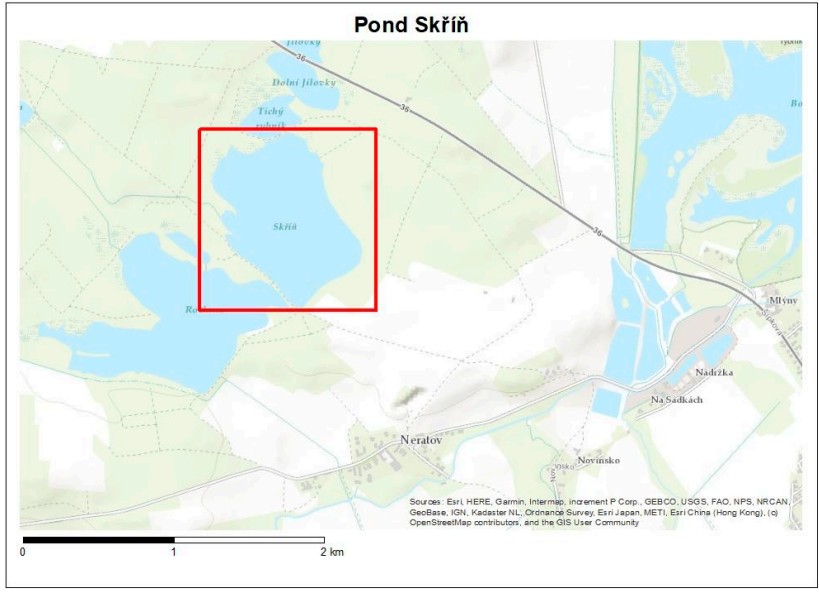

**Figure 3.** Pond Skříň, data source: [42]. The red box bounds the area of interest.

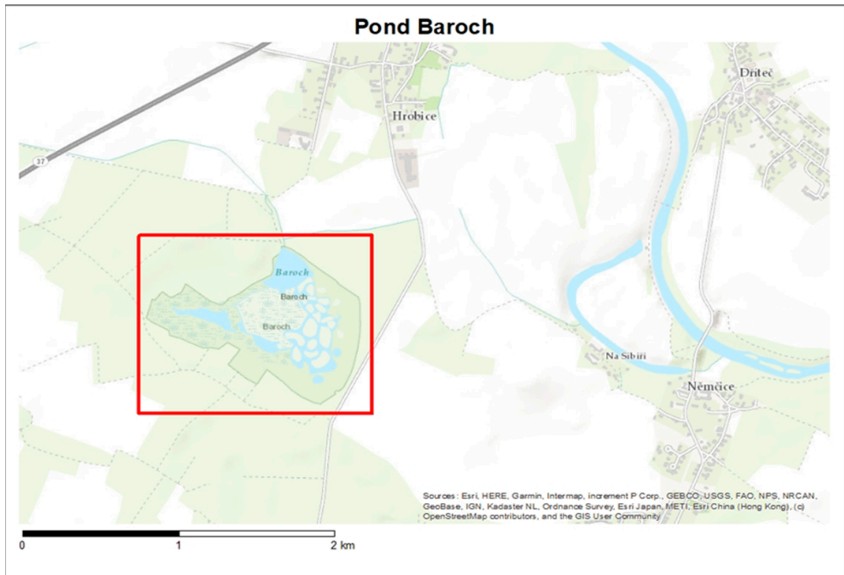

**Figure 4.** Reservation Baroch, data source: [42]. The red box bounds the area of interest.

## 4. Data Collection

Phantom 3 Pro was used for the data collection. The characteristics of this UAV are as follows: weight, 1216 g, four motors, max. flight time, 25 min. The flight with UAV is performed under suitable weather conditions. The drone contains a built-in ultra HD camera. The camera has an F2.8 lens with a 20 mm focal length and a viewing angle of 94 degrees. The built-in camera is attached to the drone by a three-axis gimbal. The drone itself costs approx. 700–800 Eur; this can be taken as a low-cost solution [43].

The planned flight was used for comprehensive data collection. The planned flight ensures low data redundancy, selects the optimal points for capturing images (waypoints), and ensures a smooth flight of the desired area. The flight was planned in DJI software, DJI GS PRO for iPad (version 2.0.16), and according to the rules for obtaining high-quality data [43]. The data from the planned flight are collected in the form of individual images with a 60% overlap with each other, a 60 m altitude, and a 90-degree angle of view (perpendicular to the drone); see Figure 5. In addition, the flight itself is subject to legislation on the operation of unmanned aerial vehicles. The rules on flying are administered by the CAA, called ÚCL Czechia [44].

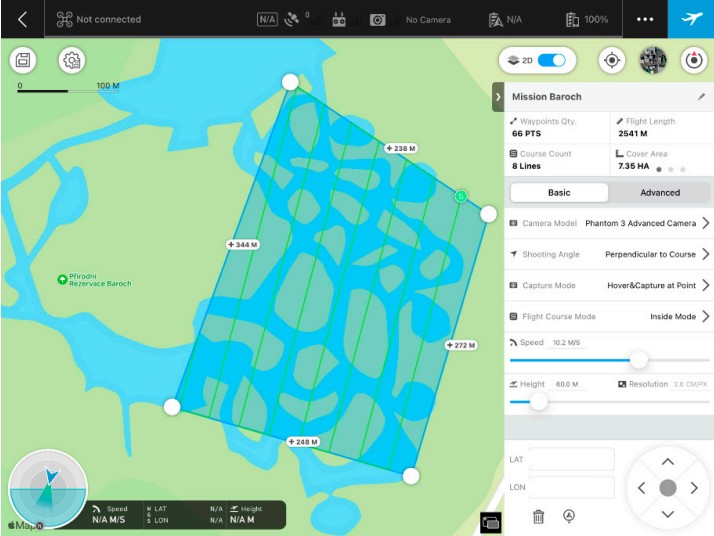

**Figure 5.** Screen from the flight planning software, source: authors.

## 5. Data Preprocessing and Processing

After obtaining the data, the next step is preprocessing. The individual data (images) must be composed into a mosaic. For mosaicking data, Icaros OneButton was used. Atmospheric correction does not need to be performed because the flight is performed at a low altitude and under very similar lighting conditions (same light conditions—sunny, beginning of the same month, around 10 am). Figures 6 and 7 represent the selected frame of individuals' areas of interest with 2.6 cm $\times$ px$^{-1}$. Processing was based on the default setting of methods with some modifications; see Table 3.

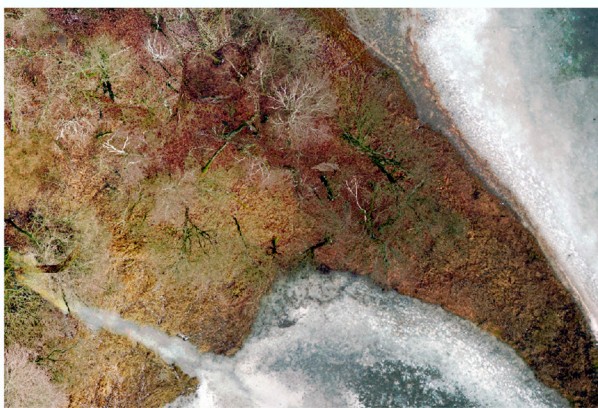

**Figure 6.** Area of interest—Skříň.

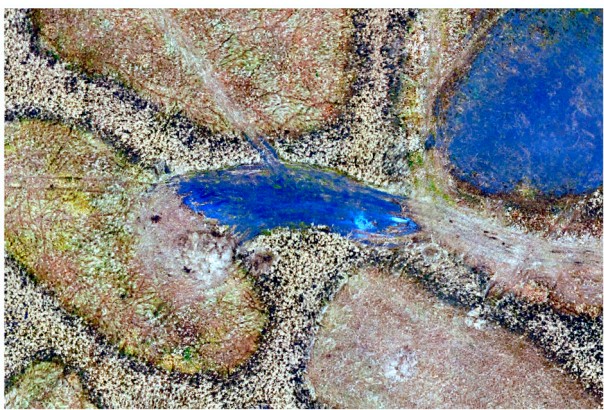

**Figure 7.** Area of interest—Baroch.

**Table 3.** Parameters of methods.

| Method | Parameters |
|---|---|
| Manual Identification | Manual vectorization of areas into 2 classes |
| Iso Cluster | Classification into 2 classes; 20 iterations |
| Maximum Likelihood | Train set based on manual selection of each class; 5 samples per class |
| Random Trees | Same train set as Maximum Likelihood; number of trees: 50; tree depth: 30; maximum number of samples per class 1000 |
| SVM | Same train set as Maximum Likelihood; number of samples per class: 500 |
| Deep Learning | Same train set as Maximum Likelihood; U-net (convolutional neural network) pixel classification; ResNet-34 (convolutional neural network with 34 layers) as Backbone model; 10 epochs |

## 6. Results and Discussion

ArcGIS PRO was used for the calculations for all the methods. ArcGIS for Desktop was used to perform the accuracy assessment and data visualization. The calculations of the Kappa index were carried out in Supplementary File S1 to make the calculation easier.

The following methods were used: Iso cluster, Maximum Likelihood, random trees, support vector machine (SVM) and deep learning (supported by ArcGIS PRO version 2.6). Both collected images were used for image classification by selected the methods. The methods were set to classify land covers into two classes, only allowing their comparison from the point of view of ice surface differentiation from all other land covers. Thus, the chosen classes were ice and all other covers. Generally, supervised methods provide better results, but it highly depends on the selected training sets. In the case of Skříň, deep learning was the best classifier, with a 92% classification success, see Figure 8. The worst classification method was the Iso cluster, with a 69% classification success, see Figure 9. In the case of Baroch, Maximum Likelihood was the best classifier, with a 93% classification success. The worst classifier was the Iso cluster, with an 88% classification success.

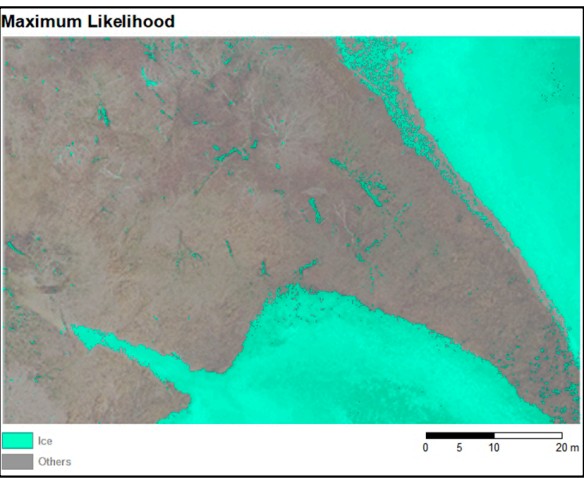

**Figure 8.** Skříň, the best classification.

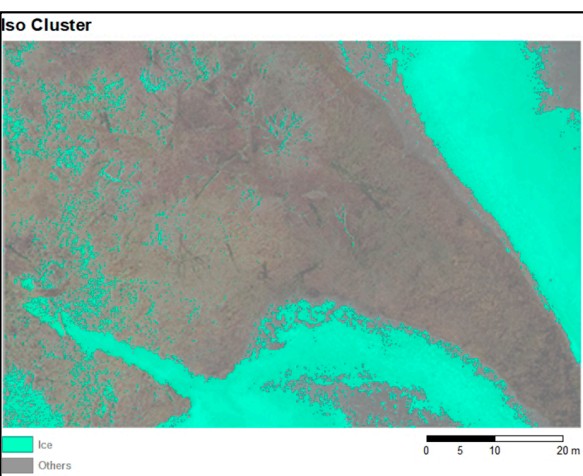

**Figure 9.** Skříň, the worst classification.

There was a problem with the classification of the edge of islands in the case of the Baroch image, see Figure 10. The edge consisted of frozen vegetation and pieces of pure ice. In the RGB spectre, the combination looks like pure ice and has almost the same pixel colour as ice. In this case, the worst method had a slightly worse result.

Table 4 shows the results of all the classification methods. The support vector machine can be considered as the best method on average. The SVM provided the second-best results in both areas, with an accuracy of around 90%.

Results of all classifications for both areas of interest are visualized, see Figure 11.

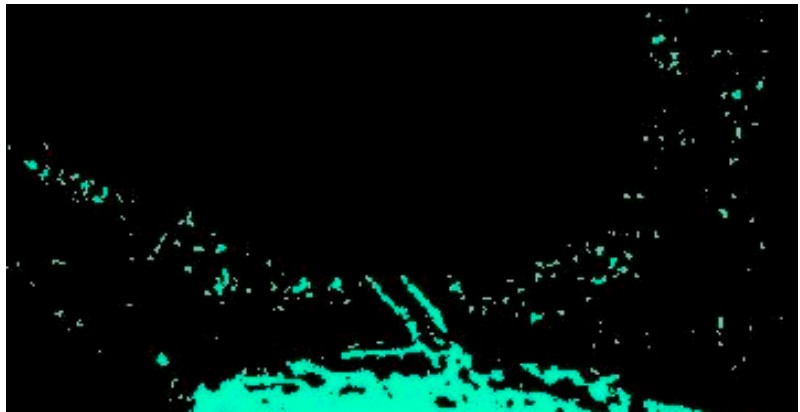

**Figure 10.** Detail of Baroch's frozen edges.

**Table 4.** Classification results—Kappa coefficient.

| Method | Pond Skříň | Pond Baroch |
|---|---|---|
| Manual identification—reference data | 1 | 1 |
| Iso Cluster | 0.6946 | 0.8842 |
| Maximum Likelihood | 0.8464 | 0.9312 |
| Random Trees | 0.8175 | 0.9175 |
| Support Vector Machine | 0.8940 | 0.9225 |
| Deep Learning | 0.9212 | 0.8889 |

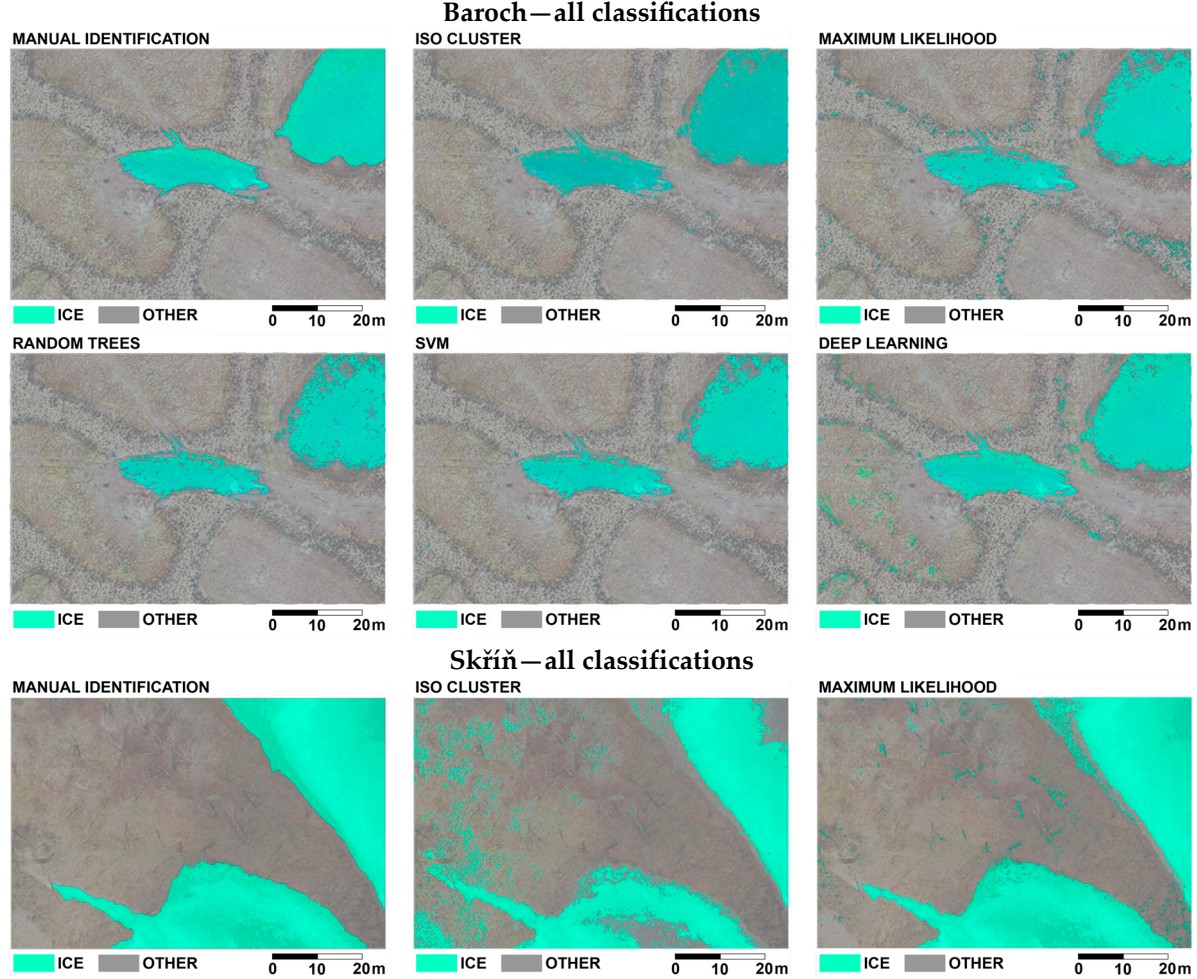

**Figure 11.** *Cont.*

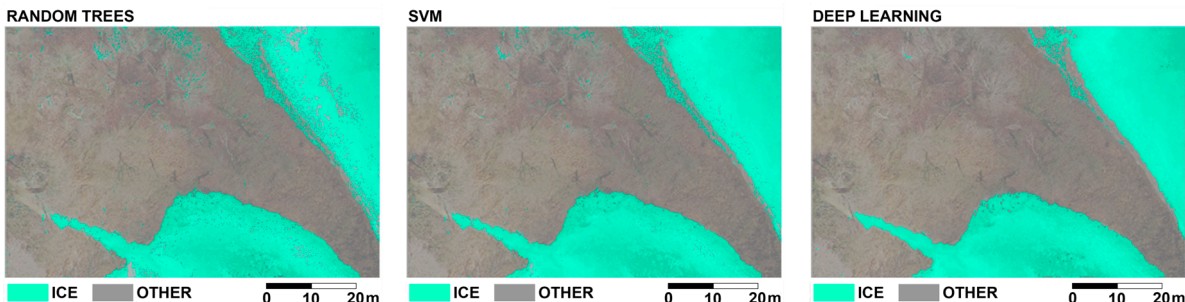

**Figure 11.** Results of all classifications for both areas of interest.

The results are in agreement with other authors. Overall, the supervised methods provide better results than the unsupervised methods. The classification accuracy of the supervised methods is around 90%, in line with the article [34]. Remarkably, the deep learning method provided the highest level of accuracy in our study. This result aligns with other authors [37,40]. The unsupervised method provided worse results than supervised methods, and its accuracy was around 70%. On the other hand, unsupervised methods can provide similar results to supervised methods in specific cases of input data (e.g., homogeneous areas).

## 7. Conclusions and Future Work

Today, it is essential to classify particular land cover types to support environmental protection, water management, and sustainable development. This article is focused on UAV-borne very-high-spatial-resolution monitoring of the ice surface.

UAVs can provide very-high-spatial-resolution data on demand, of course, with respect to the legal and weather conditions. On the other hand, cheaper UAVs may be equipped with a standard RGB camera only. In this case, not all standard remote sensing methods can be used for data processing; e.g., many spectral indices require other bands; RGB bands are insufficient.

The article provides a comparison of various pixel-based classifications, which were used to classify RGB imagery. The classification aimed to identify and distinguish the ice surface from the other land cover types.

The direct focus on identifying only ice surfaces represents a fundamental limitation of this study. We did not distinguish between different types of ice (e.g., dry and wet). RGB bands do not allow the utilization of many spectral indices and the full range of the reflectance curve.

Future work can expand on this work by including object-based classification methods, e.g., OBIA or an object-based version of deep learning, and comparing these methods.

Next, there is space to improve the deep learning training phase to prevent classification mistakes, e.g., based on frozen edges.

**Supplementary Materials:** The following supporting information can be downloaded at: https://www.mdpi.com/article/10.3390/app132011400/s1, File S1: The results of searches in the WoS and Scopus databases during the literature review phase.

**Author Contributions:** Conceptualization, J.K.; Methodology, J.J. and D.B.; Software, J.J.; Validation, J.J. and D.B.; Formal Analysis, J.J.; Investigation, J.J.; Resources, J.J. and D.B.; Data Curation, J.J.; Writing—Original Draft Preparation, J.J. and J.K.; Writing—Review and Editing, J.K. and D.B.; Visualization, J.J.; Supervision, J.K.; Project Administration, J.K. All authors have read and agreed to the published version of the manuscript.

**Funding:** The authors disclosed receipt of the following financial support for the research, authorship, and/or publication of this article: this article was supported by the Student Grant Competition of the University of Pardubice (grant No. SGS_2023_013).

**Institutional Review Board Statement:** Not applicable.

**Informed Consent Statement:** Not applicable.

**Data Availability Statement:** The results of searches in the WoS and Scopus databases during the literature review phase are available in the attached Supplementary File S1.

**Conflicts of Interest:** The authors declare no conflict of interest.

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
