# Peer review of "Comparison of Automatic Classification Methods for Identification of Ice Surfaces from Unmanned-Aerial-Vehicle-Borne RGB Imagery"

_applsci, doi:10.3390/app132011400_

Round 1

Reviewer 1 Report

The expansion of UAVs in the image classification domain is elucidated in this paper. Through an example of ice versus other land cover type classifications, and an example of Pond SkÅ™íň and the Baroch Nature Reserve, this work can add more value to the converging areas of drones and remote sensing if the following issues are addressed:

1. Make the introductory section more focussed and relevant with information about similar applications and research studies. Put one explanatory infographic diagram.

2. Provide Excel sheets for the data sources of Figures 1 and 2. In Fig 2, see if more information is presented by using the ink and surface efficiently. Also, mention the query information and data of access.

3. Create a table of classifications and chronology for various techniques and their evolution respectively (page 6).

4. Make a flowchart for data collection, pre-processing, and methodology section. Bring in more references to cite your work like the application of remote sensing in troubled areas, UN peacekeeping, fisheries, forestry etc.

5. Re-write the conclusion and future work by relating it with the research objectives stated in the opening sections.

Try to improve the manuscript's language, restructure the paragraphs and avoid repetition of arguments until anything new is referenced. 

Author Response

Dear reviewer,

Thank you very much for your valuable feedback. Please see the attached file for our response. The corresponding revisions/corrections are highlighted in the re-submitted file (we used yellow colour).

Kind regards

Authors

Reviewer 2 Report

The paper describes a comparative analysis of pixel-based classification methods for distinguishing ice from other land cover types in high-resolution RGB imagery. The article demonstrates a comprehensive evaluation of classification techniques by comparing both traditional approaches (ISODATA, Maximum Likelihood) and modern methods (SVM, random forest, deep learning). The results indicate that deep learning and Maximum Likelihood achieved the highest classification accuracy, surpassing 92% in the first area of interest. Additionally, the identification of SVM as the best classifier on average for both areas provides valuable insights into the overall performance of the methods across different scenarios. Below are some my concern:

·         The linguistic of the paper can be further improved. There are some typos. Some comma and spaces between words are missed. The language used throughout the manuscript needs to be improved

·         The authors should provide more details on how newcomers in the field, such as research students, can utilize the current methods for new cases.

·         If possible improved the discussion on the limitations and challenges associated with each classification method.

·         The author should supplement the deep learning related literature on 2023 and need to add more works that are relevant.  The Facial Expression Recognition Using Deep Neural Network,  A. Modified Genetic Algorithm with Deep Learning for Fraud Transactions of Ethereum Smart Contract, A Machine Learning based Approach to Detect the Ethereum Fraud Transactions with Limited Attributes.

·         The paper would be improved by addressing the factors such as computational requirements, training data availability, and potential biases.

Moderate editing of English language required

Author Response

(The authors gave the same response as above.)

Reviewer 3 Report

The purpose of the manuscript is to compare and assess the performance of different classifiers for ice detection using UAV images. This article is very relevant to the broad audience of the Journal of Applied Sciences. However, the manuscript is messy and needs deep restructuring and editing. The major issue is terminology. The authors have used various terms that are not correct and may be due to the issue of translation. Additionally, there is a lack of coherence. I believe that the authors should focus on performance and remove the bibliographical analysis from the introduction. I think both topics are not well addressed in this manuscript. I am not an English native speaker, but the grammar certainly needs some improvement.  Lastly and most importantly, the issue of ground data and performance indicators (e.g., R2, RMSE,  ..etc.) was not discussed. Generating binary maps should not be the end of this; authors need further steps. Additionally, the number of classes is very important when we analyse the classifiers' performance. The authors need to increase the number of classes, for example, dry ice, wet ice, fresh ice, vegetation, bare soil...

Some specific comments:

Define SVM in the abstract.

Keywords: try not to repeat title words because the title is keywords (e.g., remove image classification; automatic classification; RGB; RGB image data; UAV; supervised classification; unsupervised classification). SVM in words.

Page 1: “accurate agriculture” Do you mean precision agriculture?

Remove the bibliographical analysis from the introduction.

All figures and table captions should be reconsidered. The captions are not clear and are not appropriate. Write descriptive but concise captions that tell the reader the full idea about the table and figure. For example, in Table 1 what are records? Figure 2 What is UA? Figure 3, not a proper map? Where is this?

Page 11 “The flight was planned in software from DJI and planned” Which software?

60% overlay Do you mean overlap/side lap?

A screenshot of the flight plan will be good. How many images have the orthomosiacs been developed? What was the accuracy?  What about GCPS? Ground-truthing?

Figure shows the sample distribution.

The terminologies are not correct. 

Author Response

(The authors gave the same response as above.)

Round 2

Reviewer 3 Report

A proper map for Figures 3 &4 should be provided for more clarity for the international audience. 

All figure captions need to be improved and be concisely descriptive. e.g.  "Results of all classifications" add the name of all classifiers with indicators a, b, c...